# Epidemiology of *Nocardia* Species at a Tertiary Hospital in Southern Taiwan, 2012 to 2020: MLSA Phylogeny and Antimicrobial Susceptibility

**DOI:** 10.3390/antibiotics11101438

**Published:** 2022-10-19

**Authors:** Shu-Fang Kuo, Fang-Ju Chen, I-Chia Lan, Chun-Chih Chien, Chen-Hsiang Lee

**Affiliations:** 1Department of Laboratory Medicine, Kaohsiung Chang Gung Memorial Hospital, Kaohsiung 183301, Taiwan; 2Department of Laboratory Medicine, Chiayi Chang Gung Memorial Hospital, Chiayi 261363, Taiwan; 3Department of Medical Biotechnology and Laboratory Sciences, College of Medicine, Chang Gung University, Taoyuan 333302, Taiwan; 4Division of Infectious Diseases, Department of Internal Medicine, Kaohsiung Chang Gung Memorial Hospital, Kaohsiung 483301, Taiwan; 5Division of Infectious Diseases, Department of Internal Medicine, Chiayi Chang Gung Memorial Hospital, Chiayi 561363, Taiwan; 6School of Medicine, College of Medicine, Chang Gung University, Taoyuan 633302, Taiwan

**Keywords:** nocardiosis, multi-locus sequence analysis, phylogenetic tree analysis, trimethoprim/sulfamethoxazole, imipenem

## Abstract

The identification and antimicrobial susceptibility of *Nocardia* spp. are essential for guiding antibiotic treatment. We investigated the species distribution and evaluated the antimicrobial susceptibility of *Nocardia* species collected in southern Taiwan from 2012 to 2020. A total of 77 *Nocardia* isolates were collected and identified to the species level using multi-locus sequence analysis (MLSA). The susceptibilities to 15 antibiotics for *Nocardia* isolates were determined by the broth microdilution method, and the MIC_50_ and MIC_90_ for each antibiotic against different species were analyzed. *N. cyriacigeorgica* was the leading isolate, accounting for 32.5% of all *Nocardia* isolates, and the prevalence of *Nocardia* isolates decreased in summer. All of the isolates were susceptible to trimethoprim/sulfamethoxazole, amikacin, and linezolid, whereas 90.9% were non-susceptible to cefepime and imipenem. The phylogenic tree by MLSA showed that the similarity between *N. beijingensis* and *N. asiatica* was as high as 99%, 73% between *N. niigatensis* and *N. crassostreae*, and 86% between *N. cerradoensis* and *N. cyriacigeorgica*. While trimethoprim/sulfamethoxazole, amikacin, and linezolid remained fully active against all of the *Nocardia* isolates tested, 90.9% of the isolates were non-susceptible to cefepime and imipenem.

## 1. Introduction

Nocardiosis is caused by several species of *Nocardia*, a ubiquitous bacterium in the environment that is transmitted by inhalation or direct cutaneous inoculation [1]. *Nocardia* species are aerobic, partially acid-fast, beaded, branched Gram-positive bacilli with colonies of filamentous, slow-growing, soil-borne bacteria [1,2]. *Nocardia* spp. is responsible for a variety of clinical infections, ranging from skin and soft tissue infections to respiratory and central nervous system infections [3]. Monitoring the epidemiological characteristics of nocardiosis including species distribution, clinical features, and antimicrobial susceptibility profiles is warranted to inform diagnostic and treatment decisions [4].

Different *Nocardia* species may have different geographic distributions, pathogenic characteristics, and antimicrobial susceptibility patterns [5]. Pulmonary nocardiosis usually leads to high mortality and morbidity if not diagnosed in time to initiate the appropriate antimicrobial treatment [6]. Therefore, the identification of *Nocardia* isolates at the species level and the determination of their antimicrobial susceptibility are critical for the delivery of appropriate patient care [7].

This study aimed to investigate the species distribution and evaluate the antimicrobial susceptibility patterns of individual *Nocardia* species isolated from patients seeking care at a referral hospital in southern Taiwan from 2012 to 2020. Speciation of *Nocardia* isolates was performed using multi-locus sequence analysis (MLSA).

## 2. Results

### 2.1. Patient Characteristics

During the 9-year study period, a total of 77 patients were diagnosed with nocardiosis, 63.6% of whom were male and had an age ranging from 31 to 97 years. The clinical characteristics of the 77 *Nocardia* isolates are shown in Table 1.

### 2.2. Distribution of Nocardia Species

Of the 77 *Nocardia* isolates, 12 type strains were identified under a phylogenetic tree constructed from the concatenated *gyrB*-16S rRNA-*secA1*-*hsp65* sequences. *N. cyriacigeorgica* was the most common species (*n* = 25, 32.5%), followed by *N. farcinica* (*n* = 18, 23.4%), *N. brasiliensis* (*n* = 13, 16.9%), *N. beijingensis* (*n* = 9, 11.7%), *N. asiatica* (*n* = 3, 3.9%), *N. asteroides* and *N. concava* (*n* = 2, 2.6%), *N. amikacinitolerans*, *N. cerradoensis*, *N. crassostreae*, *N. niigatensis*, and *N. otitidiscaviarum* (*n* = 1, 1.3%).

The correlations between the drug susceptibility patterns and *Nocardia* species are shown in Table 2. The most common drug pattern was type V to type VIII (74.1%). In contrast to the drug pattern types described previously by McTaggart et al. [8], we found that both the *N. farcinica* and *N. cyriacigeorgica* strains were IPM-resistant and the *N. cyriacigeorgica* strains were also FEP-resistant.

### 2.3. Nocardia Species Profile by Analysis of Years and Months

In 2012, 2014, 2019, and 2020, the predominant species was *N. cyriacigeorgica* (Figure 1). In contrast, the predominant species identified in 2013, 2016, 2017, and 2018 was *N. farcinica*. *N. brasiliensis* was the predominant species in 2015. Figure 2 shows the monthly distribution of the *Nocardia* species, suggesting that the prevalence of *Nocardia* infections was lower in summer and higher in autumn.

### 2.4. Antibiotic Susceptibility Profiles

The MIC_50_ and MIC_90_ values (in µg/mL) and the MIC ranges and distributions for each *Nocardia* species are shown in Table 3. All *Nocardia* isolates in our study were susceptible to SXT, AN, and LZD. Of these isolates, 23.4% were non-susceptible to TOB. In contrast, 90.9% of *Nocardia* isolates were not susceptible to FEP and IPM, especially all isolates of *N. cyriacigeorgica*, *N. farcinica*, and *N. brasiliensis*. We also found that 83.1% of the isolated strains were non-susceptible to CLR, and 80.5% were non-susceptible to CIP. The susceptibility breakpoints for tigecycline and cefoxitin were not established.

### 2.5. PFGE for N. cyriacigeorgica

*N. cyriacigeorgica* was the most common species in this study. We randomly selected nine strains of *N. cyriacigeorgica*, which were isolated in 2019 and 2020 for PFGE analysis to determine the genetic relatedness among the strains. Figure 3 shows that all nine strains were isolated from different patients, and their parental similarities were less than 60%, indicating non-homologous strains.

### 2.6. Phylogenetic Tree by MLSA Scheme

In our study, there were 12 *Nocardia* spp. The differences in four-locus (*gyrB*-16S rRNA-*secA1*-*hsp65*) MLSA concatenated sequences among these 12 species and the evolutionary phylogenetic trees are shown in Figure 4. The similarity between *N. beijingensis* and *N. asiatica* was as high as 99%, and that between the two species and *N. farcinica* was 77%. *N. beijingensis* and *N. asiatica* belonged to the *N. abscessus* complex [7], and their nucleic acid similarity was up to 99%. *N. niigatensis* and *N. crassostreae* had a similarity of 73%. *N. cerradoensis* had an 86% similarity to *N. cyriacigeorgica*. Additionally, *N. amikacinitolerans* was independent and had greater differences than the other species.

## 3. Discussion

Recently, MALDI-TOF MS has been shown to provide an accurate identification of *Nocardia* species when an augmented *Nocardia* library is employed. However, while some species are easily identified (i.e., *N. brasiliensis*), for others, the identification has only been shown to extend to the complex level (*N. abscessus* complex, *N. brevicatena-N. paucivorans* complex, *N. nova* complex, and *N. transvalensis* complex). The identification of uncommon species remains a challenge [7,9]. Sequence analysis of the 16S rRNA gene is suggested as the “gold standard” for the identification of *Nocardia* isolates to the species level. However, when the identification to species level is based on the partial 5’ 16S rRNA sequencing, as in this case, a second genetic locus such as the *secA1* gene for isolate identification is recommended because 16S rRNA sequence analysis alone provides insufficient species-level resolution for many *Nocardia* spp., whereas *secA1* gene sequence analysis is more discriminatory and gives better resolution to the species level [10]. Both genes were included in the MLSA schema employed in this study for the species assignation to achieve higher accuracy and differentiation. Nevertheless, the identification of *Nocardia* isolates in some challenging species, species groups, or complexes is not possible with MLSA. In our study, there was a 99% similarity between *N. beijingensis* and *N. asiatica* in the MLSA analysis, and all of them belonged to the *N. abscessus* complex, which was difficult to distinguish.

Previous studies conducted before 2010 indicated that the most common *Nocardia* spp. in Taiwan was *N. brasiliensis* [11,12]. In contrast, *N. farcinica* was the most common isolated species in China from 2009 to 2021 [13]. *N. nova* complex organisms were the most common isolates in the United States before 2004 and Canada before 2008 [14,15], and *N. cyriacigeorgica* was the most common pathogen in Spain before 2008 [16]. *N. cyriacigeorgica* was the most common causative agent of pulmonary nocardiosis in southern Taiwan from 2004 to 2010 [17] and China from 2010 to 2020, where pulmonary nocardiosis (90.2%) was the most common clinical presentation of infection [18], which is consistent with our study predominated in lung infection (Table 1) conducted between 2012 and 2020. There are few recent epidemiological data on invasive nocardiosis in this region. Further studies are required to confirm whether *N. cyriacigeorgica* is an emerging pathogen in southern Taiwan.

The different species of *Nocardia* isolates exhibit diverse susceptibilities to antibiotics. Our study showed that all *Nocardia* spp. are susceptible to SXT, LZD, and AN. In contrast, the non-susceptibility rates of *Nocardia* spp. to DOX and MIN were 80.5% and 71.4%, respectively. Overall, SXT, LZD, and AN were the most active drugs for all *Nocardia* spp., which is consistent with the findings of other studies [8,10,19]. Our study showed that all *Nocardia* spp. were susceptible to TOB, except for *N. farcinica*. This suggests that TOB should be avoided in infections with *N. farcinica* in our region.

Although our study showed that most of the drug patterns were consistent with the drug pattern types suggested by McTaggart [8], different antibiograms were found in the current study. In agreement with the report of Tan et al. [10], high IPM resistance rates were observed in both *N. farcinica* and *N. cyriacigeorgica*, and high FEP resistance rates were observed in *N. cyriacigeorgica* in our study (Table 3). High IPM resistance in *N. cyriacigeorgica* was observed in Australia [20], but not in Spain [19] and Canada [8]. Another study of 151 *Nocardia* isolates conducted in four hospitals in Taiwan between 1998 and 2009 found that the three leading *Nocardia* spp. were *N. brasiliensis*, *N. cyriacigeorgica*, and *N. farcinica*. The susceptibility of *N. brasiliensis*, *N. cyriacigeorgica*, and *N. farcinica* to IPM was 47%, 100%, and 100%, respectively [11]. The higher rate of non-susceptibility of IPM observed in our study could either be a unique regional resistance profile of *Nocardia* spp. in southern Taiwan or selection pressure from the overuse of carbapenems [21,22]. This finding suggests that FEP and IPM should not be used empirically until the antimicrobial susceptibility results are available. Further epidemiological surveillance of the antimicrobial susceptibility profiles of *Nocardia* spp. is warranted to confirm our findings.

Nocardiosis occurs worldwide. *Nocardia* infections have increased in the past decades, likely due to improved detection and identification methods and the expanding immunocompromised population [3]. Although reports of community-acquired nocardiosis are common, few cases of nosocomial transmission of *Nocardia* species have been reported [23,24,25,26]. *N. cyriacigeorgica* has also been reported to cause outbreaks [27]. We performed a PFGE analysis for *N. cyriacigeorgica*, and the genetic relatedness of the strains from different patients were not homologous (Figure 3). Remarkably, no outbreaks occurred in this study. Our finding of a decrease in the prevalence of clinically isolated *Nocardia* spp. in summer from 2012 to 2020 is in contrast to the findings of an Australian environmental survey of *Nocardia* species isolated during a 1-year period from the foaming marine waters of the Sunshine Coast region [28], which suggests that hot weather is conducive to the growth of *Nocardia*. However, more studies of the prevalence of *Nocardia* species among clinical samples per month are needed to gain insights into the correlation of climate change and the distribution of *Nocardia* spp.

Our study had some limitations. First, the number of isolated *Nocardia* spp. in this study was still low, which may have prevented us from exactly determining the prevalence. Second, we did not investigate the molecular mechanisms of the antimicrobial resistance of the collected *Nocardia* strains to explain the regional differences in the antimicrobial susceptibility profiles.

## 4. Materials and Methods

### 4.1. Bacterial Isolates

Non-duplicated 77 isolates of *Nocardia* spp. collected from all patients who received a culture-confirmed diagnosis of nocardiosis at Kaohsiung Chang Gung Memorial Hospital (KCGMH) were included from 1 January 2012 to 31 December 2020. The KCGMH is a 2700-bed facility that serves as a primary care and tertiary referral center in southern Taiwan.

### 4.2. Housekeeping Gene Selection, DNA Extraction, PCR, and Sequencing

According to previous studies, four housekeeping genes (16S rRNA, *secA1*, *gyrB*, and *hsp65*) were selected [29,30]. DNA was extracted using a QIAGEN DNeasy Tissue Kit. The PCR products were referred to the Genome Sequencing Company for sequencing. The gene sequences were subsequently matched to those in the National Center for Biotechnology Information database (http://www.ncbi.nlm.nih.gov, accessed on 21 November 2021) to identify the *Nocardia* species [14,29]. The gene sequences were deposited in the GenBank database and their corresponding accession numbers are presented in Appendix A.

### 4.3. Construction of Phylogenetic Tree

MLSA using concatenated sequences of *gyrB*-16S-*secA1*-*hsp65* has previously been used to identify *Nocardia* species [29,30]. Primer sequences published by McTaggart et al. [31] are presented in Appendix A. Phylogenetic trees were constructed using the neighbor-joining method (software: Molecular Evolutionary Genetics Analysis across Computing Platforms). Bootstrap values based on 1000 replications were listed as percentages at the branching points of the tree [32]. Phylogenetic trees were constructed using the neighbor-joining (NJ) genetic distance method [33] and performed using the ClustalW algorithm in the mega X software. The reliability of each tree topology was checked using 10,000 bootstrap replications [32,34].

### 4.4. Pulsed-Field Gel Electrophoresis (PFGE) Analysis

*N. cyriacigeorgica* was the most common species in this study, so we performed PFGE to clarify whether there was a possibility of nosocomial infection. The suspension (300 µL) and lysozyme (20 µL; 25 mg/mL) were added and incubated at 37 °C for 4 h after mixing. Total genomic DNA was prepared in agarose plugs and lysed in 5 mL of lysis buffer (25 mg lysozyme per mL and 20 µL proteinase K in TE buffer) for 4 h in a 56 °C water bath. The plugs were digested with XbaI. DNA fragments were separated on a 1% gel in a CHEF Mapper System (Bio-Rad, Mississauga, Ontario, Canada) with linear ramping pulse times of 1–30 s over 17.5, 6 V/cm at 14 °C. The Dice coefficients of the PFGE profiles were analyzed with an UPGMA dendrogram using GelCompar II version 6.6.11 (Applied Maths BVBA, Kortrijk, Belgium).

### 4.5. Antimicrobial Susceptibility Test

The susceptibility of the isolates to 15 commonly-used antibiotics was tested by the microbroth dilution method using Sensititre RAPMYCO TREK (Sensititre Susceptibility plates; TREK Diagnostic Systems Ltd. Cleveland, OH, USA) according to the manufacturer’s instructions. The strains recommended by the CLSI, *S. aureus* ATCC 29213 and *E. coli* ATCC 25922, were tested for quality control.

Antibiotics chosen for susceptibility testing in this study included amikacin (AN), amoxicillin/clavulanic acid (AMC), cefepime (FEP), cefoxitin (FOX), ceftriaxone (CRO), ciprofloxacin (CIP), clarithromycin (CLR), doxycycline (DOX), imipenem (IPM), linezolid (LZD), minocycline (MIN), moxifloxacin (MXF), tigecycline (TGC), tobramycin (TOB), and trimethoprim/sulfamethoxazole (SXT). The results were interpreted according to CLSI guideline M62 for aerobic actinomycetes [35].

### 4.6. Antimicrobial Susceptibility Patterns

According to Wallace et al. [36], six patterns of antibiotic susceptibility to *Nocardia* spp. have been proposed. These include *N. abscessus* complex (drug pattern I) and *N. brevicatena*/*N. paucivorans* (drug pattern II), *Nocardia nova* complex (drug pattern III), *Nocardia transvalensis* complex (drug pattern IV), *N. farcinica* (drug pattern V), and *N. cyriacigeorgica* (drug pattern VI) [36,37]. McTaggart et al. [8] suggested numerous rarely-occurring species using broth microdilution and divided them into four other drug patterns. We also characterized the antimicrobial resistance of several *Nocardia* isolates and profiled their antimicrobial susceptibility patterns.

## 5. Conclusions

*N. cyriacigeorgica* was the major *Nocardia* spp. identified in this study. SXT, LZD, and AN were the most active antimicrobial agents against all *Nocardia* strains identified. The distribution and antibiotic resistance characteristics of *Nocardia* species further our understanding of the diversity of circulating *Nocardia* species and inform the decision-making in the choice of empirical therapy.

## Figures and Tables

**Figure 1 antibiotics-11-01438-f001:**
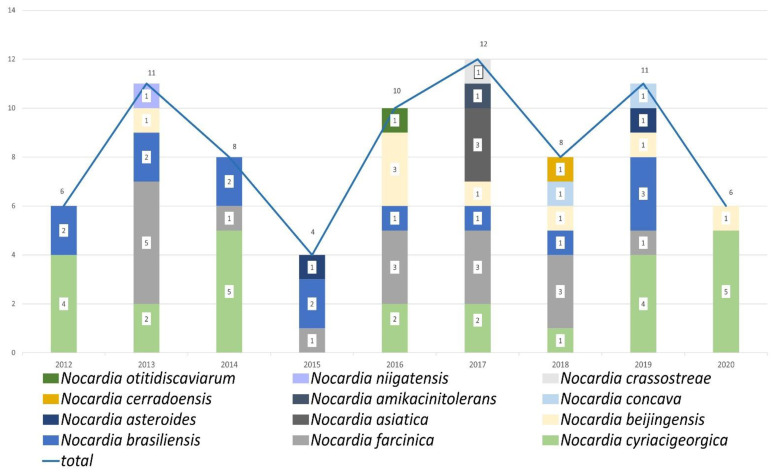
*Nocardia* species distribution by year of isolation.

**Figure 2 antibiotics-11-01438-f002:**
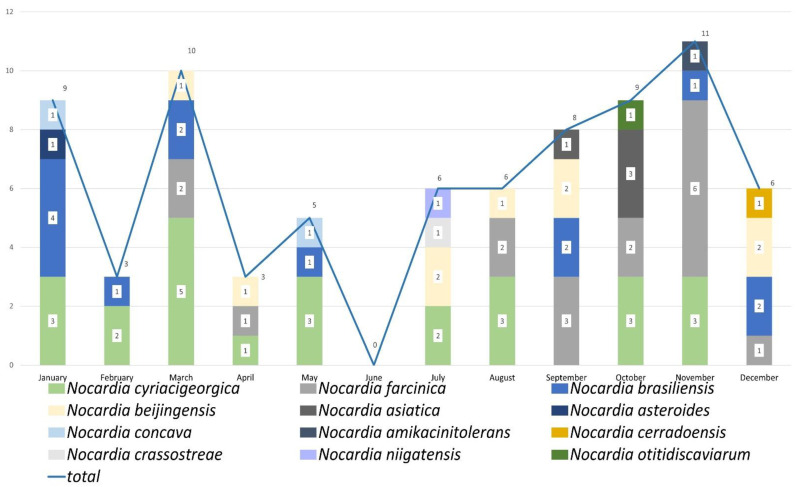
*Nocardia* species distribution by month of isolation.

**Figure 3 antibiotics-11-01438-f003:**
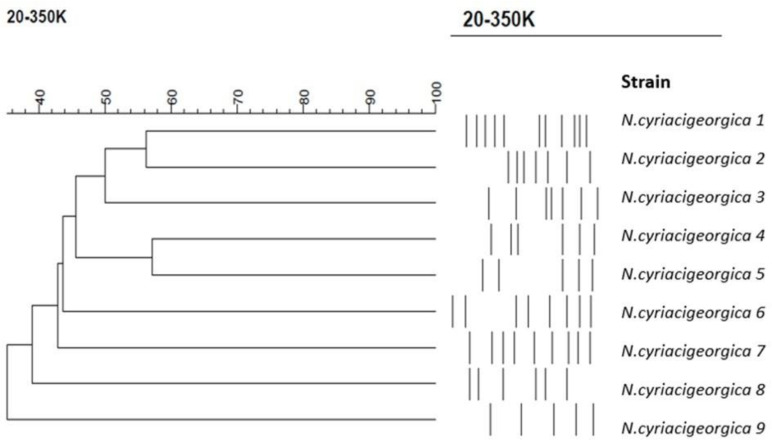
Genetic relationships of *N. cyriacigeorgica* by PFGE analysis.

**Figure 4 antibiotics-11-01438-f004:**
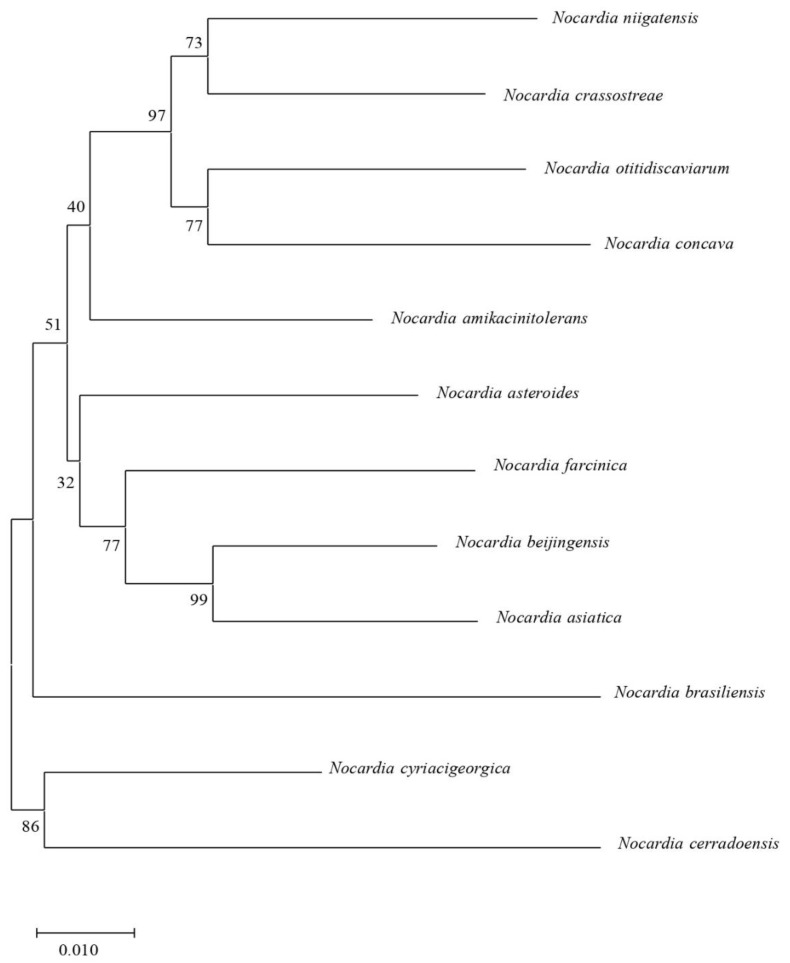
A phylogenetic neighbor-joining tree including these 12 types of strains as an indicator and the 77 clinical strains studied in the nine years based on the MLSA concatenated sequence.

**Table 1 antibiotics-11-01438-t001:** The clinical characteristics of the 77 included patients.

Characteristics	
Gender, *n* (%)	
Male	49 (63.6)
Age (years)	
Median (range)	76 (31–97)
Mean ± standard deviation	70.4 ± 15.7
Specimen type, *n* (%)	
Pus	21 (27.3)
Sputum	14 (18.2)
Wound	11 (14.3)
Blood	7 (9.1)
Abscess	5 (6.5)
Bronchial washing	7 (9.1)
Corneal ulcer	4 (5.2)
Pleural effusion	4 (5.2)
Synovial fluid	2 (2.6)
Cerebrospinal fluid	1 (1.3)
Bone tissue	1 (1.3)
Site of involvement, *n* (%)	
Lung	35 (45.5)
Central nervous system	9 (11.7)
Skin and soft tissue	19 (24.7)
Bone and joint	7 (9.1)
Blood stream	7 (9.1)
Disseminated (including blood stream)	18 (23.3)

**Table 2 antibiotics-11-01438-t002:** The antimicrobial susceptibility patterns of different *Nocardia* species.

*Nocardia* Species	No. of Isolates	Drug Patterns Types	Antimicrobial Susceptibility Pattern	
			Non-Susceptible (%)	Susceptible (%)
*N. farcinica*	18	V	IPM (100)	SXT (100)
			FEP (100)	LZD (100)
			DOX (100)	AN (100)
			TOB (100)	
			CLR (100)	
*N. cyriacigeorgica*	25	VI	CIP (100)	SXT (100)
			IPM (100)	LZD (100)
			MXF (100)	AN (100)
			FEP (100)	TOB (100)
			AMC (100)	
			CLR (92)	
*N. brasiliensis*	13	VIII	CIP (100)	SXT (100)
			IPM (100)	LZD (100)
			FEP (100)	AN (100)
			CRO (92)	TOB (100)
			DOX (100)	
			CLR (92)	
*N. otitidiscaviarium*	1	VII	CIP (100)	SXT (100)
			IPM (100)	LZD (100)
			FEP (100)	AN (100)
			AMC (100)	TOB (100)
			CRO (100)	
			CLR (100)	

Abbreviations: AN, amikacin; AMC, amoxicillin/clavulanic acid 2:1 ratio; CIP, ciprofloxacin; CLR, clarithromycin; CRO, ceftriaxone; DOX, doxycycline; FEP, cefepime; IPM, imipenem; LZD, linezolid; MIN, minocycline; MXF, moxifloxacin; SXT, trimethoprim/sulfamethoxazole; TOB, tobramycin.

**Table 3 antibiotics-11-01438-t003:** The antimicrobial susceptibility test results.

Antimicrobial Agent	Species (No. of Strains Tested)											
	*N. cyriacigeorgica* (25)	*N. brasiliensis* (13)	*N. farcinica* (18)	*N. niigatensis* (1)	*N. asteroides* (2)	*N. beijingensis* (9)	*N. otitidiscaviarum* (1)	*N. crassostreae* (1)	*N. concava* (2)	*N. cerradoensis* (1)	*N. asiatica* (3)	*N. amikacinitolerans* (1)
Trimethoprim/ Sulfamethoxazole (SXT)												
Resistant [*n* (%)]	0	0	0	0	0	0	0	0	0	0	0	0
Intermediate [*n* (%)]	0	0	0	0	0	0	0	0	0	0	0	0
Susceptible [*n* (%)]	25 (100)	13 (100)	18 (100)	1 (100)	2 (100)	9 (100)	1 (100)	1 (100)	2 (100)	1 (100)	3 (100)	1 (100)
MIC_50_ [µg/mL]	0.25/4.75	0.5/9.5	1/19			0.25/4.75					0.25/4.75	
MIC_90_ [µg/mL]	0.5/9.5	0.5/9.5	2/38			1/19					0.25/4.75	
Linezolid (LZD)												
Resistant [*n* (%)]	0	0	0	0	0	0	0	0	0	0	0	0
Intermediate [*n* (%)]	0	0	0	0	0	0	0	0	0	0	0	0
Susceptible [*n* (%)]	25 (100)	13 (100)	18 (100)	1 (100)	2 (100)	9 (100)	1 (100)	1 (100)	2 (100)	1 (100)	3 (100)	1 (100)
MIC_50_ [µg/mL]	2	2	2			1					1	
MIC_90_ [µg/mL]	2	4	4			2					1	
Ciprofloxacin (CIP)												
Resistant [*n* (%)]	25 (100)	13 (100)	4 (22.2)	0	2 (100)	5 (55.6)	1 (100)	1 (100)	1 (50)	0	3 (100)	1 (100)
Intermediate [*n* (%)]	0	0	3 (16.7)	1 (100)	0	1 (11.1)	0	0	1 (50)	0	0	0
Susceptible [*n* (%)]	0	0	11 (61.1)	0	0	3 (33.3)	0	0	0	1 (100)	0	0
MIC_50_ [µg/mL]	>4	>4	1			4					>4	
MIC_90_ [µg/mL]	>4	>4	>4			>4					>4	
Imipenem (IPM)												
Resistant [*n* (%)]	21 (84.0)	10 (76.9)	14 (77.8)	1 (100)	0	5 (55.5)	1 (100)	1 (100)	2 (100)	0	0	1 (100)
Intermediate [*n* (%)]	4 (16.0)	3 (23.1)	4 (22.2)	0	2 (100)	1 (11.2)	0	0	0	0	0	0
Susceptible [*n* (%)]	0	0	0	0	0	3 (33.3)	0	0	0	1 (100)	3 (100)	0
MIC_50_ [µg/mL]	16	32	16			16					2	
MIC_90_ [µg/mL]	16	>64	32			64					4	
Moxifloxacin (MXF)												
Resistant [*n* (%)]	24 (96.0)	0	4 (22.2)	0	1 (50)	2 (22.3)	0	0	0	0	0	1 (100)
Intermediate [*n* (%)]	1 (4.0)	11 (84.6)	0	0	1 (50)	3 (33.3)	1 (100)	0	0	1 (100)	0	0
Susceptible [*n* (%)]	0	2 (15.4)	14 (77.8)	1 (100)	0	4 (44.4)	0	1 (100)	2 (100)	0	3 (100)	0
MIC_50_ [µg/mL]	4	2	0.25			2					8	
MIC_90_ [µg/mL]	8	2	4			>8					>8	
Cefepime (FEP)												
Resistant [*n* (%)]	16 (64.0)	12 (92.3)	16 (88.8)	1 (100)	2 (100)	4 (44.4)	1 (100)	1 (100)	2 (100)	0	0	1 (100)
Intermediate [*n* (%)]	9 (36.0)	1 (7.8)	2 (11.2)	0	0	2 (22.3)	0	0	0	0	0	0
Susceptible [*n* (%)]	0	0	0	0	0	3 (33.3)	0	0	0	1 (100)	3 (100)	0
MIC_50_ [µg/mL]	32	>32	>32			16					8	
MIC_90_ [µg/mL]	>32	>32	>32			32					8	
Cefoxitin (FOX)												
MIC range	64–128	16–128	64–128	>128	16–64	8–32	>128	>128	>128	64	4–16	>64
Amoxicillin/clavulanic acid 2:1 ratio (AMC)												
Resistant [*n* (%)]	23 (92.0)	1 (7.7)	3 (16.7)	1 (100)	2 (100)	4 (44.5)	1 (100)	1 (100)	2 (100)	0	3 (100)	0
Intermediate [*n* (%)]	2 (8.0)	2 (15.4)	12 (66.6)	0	0	0	0	0	0	0	0	0
Susceptible [*n* (%)]	0	10 (76.9)	3 (16.7)	0	0	5 (55.5)	0	0	0	1 (100)	0	1 (100)
MIC_50_ [µg/mL]	32/16	8/4	16/8			8/4					>64/32	
MIC_90_ [µg/mL]	64/32	16/8	32/16			>64/32					>64/32	
Amikacin (AN)												
Resistant [*n* (%)]	0	0	0	0	0	0	0	0	0	0	0	0
Intermediate [*n* (%)]	0	0	0	0	0	0	0	0	0	0	0	0
Susceptible [*n* (%)]	25 (100)	13 (100)	18 (100)	1 (100)	2 (100)	9 (100)	1 (100)	1 (100)	2 (100)	1 (100)	3 (100)	1 (100)
MIC_50_ [µg/mL]	1	1	1			1					1	
MIC_90_ [µg/mL]	1	1	1			1					1	
Ceftriaxone (CRO)												
Resistant [*n* (%)]	2 (8.0)	10 (76.9)	15 (83.3)	1 (100)	0	1 (11.1)	1 (100)	1 (100)	2 (100)	0	0	1 (100)
Intermediate [*n* (%)]	8 (32.0)	2 (15.4)	1 (5.6)	0	0	3 (33.3)	0	0	0	0	0	0
Susceptible [*n* (%)]	15 (60.0)	1 (7.7)	2 (11.1)	0	2 (100)	5 (55.6)	0	0	0	1 (100)	3 (100)	0
MIC_50_ [µg/mL]	8	>64	>64			4					4	
MIC_90_ [µg/mL]	32	>64	>64			32					4	
Doxycycline (DOX)												
Resistant [*n* (%)]	0	1 (7.7)	1 (5.6)	1 (100)	0	0	0	0	2 (100)	0	0	0
Intermediate [*n* (%)]	17 (68.0)	12 (92.3)	17 (94.4)	0	2 (100)	6 (66.7)	1 (100)	1 (100)	0	1 (100)	0	0
Susceptible [*n* (%)]	8 (32.0)	0	0	0	0	3 (33.3)	0	0	0	0	3 (100)	1 (100)
MIC_50_ [µg/mL]	2	4	4			2					0.12	
MIC_90_ [µg/mL]	4	4	4			4					0.12	
Minocycline (MIN)												
Resistant [*n* (%)]	0	0	0	0	0	0	0	0	2 (100)	0	0	0
Intermediate [*n* (%)]	17 (68.0)	11 (84.6)	17 (94.4)	1 (100)	2 (100)	3 (33.3)	1 (100)	1 (100)	0	0	0	0
Susceptible [*n* (%)]	8 (32.0)	2 (15.4)	1 (5.6)	0	0	6 (66.7)	0	0	0	1 (100)	3 (100)	1 (100)
MIC_50_ [µg/mL]	2	4	4			1					1	
MIC_90_ [µg/mL]	4	4	4			2					1	
Tigecycline (TGC)												
MIC range	0.25–2	0.25–0.5	0.5–4	1	0.5–1	0.12–0.5	1	2	2 (100)	0.12	0.25	2
Tobramycin (TOB)												
Resistant [*n* (%)]	0	0	16 (88.9)	0	0	0	0	0	0	0	0	0
Intermediate [*n* (%)]	0	0	2 (11.1)	0	0	0	0	0	0	0	0	0
Susceptible [*n* (%)]	25 (100)	13 (100)	0	1 (100)	2 (100)	9 (100)	1 (100)	1 (100)	2 (100)	1 (100)	3 (100)	1 (100)
MIC_50_ [µg/mL]	1	1	16			1					1	
MIC_90_ [µg/mL]	1	1	>16			1					1	
Clarithromycin (CLR)												
Resistant [*n* (%)]	22 (88.0)	9 (69.2)	18 (100)	1 (100)	2 (100)	2 (22.2)	1 (100)	1 (100)	0	0	1 (33.3)	1 (100)
Intermediate [*n* (%)]	1 (4.0)	3 (23.0)	0	0	0	2 (22.2)	0	0	0	0	0	0
Susceptible [*n* (%)]	2 (8.0)	1 (7.8)	0	0	0	5 (55.6)	0	0	2 (100)	1 (100)	2 (66.7)	0
MIC_50_ [µg/mL]	>16	8	>16			1					1	
MIC_90_ [µg/mL]	>16	>16	>16			16					16	

A comparison of the activities of different antibiotics against *N. cyriacigeorgica* and *N. farcinica* revealed that the MIC_90_ values of MXF and AMC against *N. cyriacigeorgica* were higher than those against *N. farcinica*. In contrast, the MIC_90_ values of SXT, LZD, IPM, CRO, and TGC against *N. cyriacigeorgica* were lower than those against *N. farcinica* (Table 3).

## Data Availability

The datasets generated and/or analyzed during the current study are available from the corresponding author upon reasonable request.

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
