# Peer review of "Epidemiology of Nocardia Species at a Tertiary Hospital in Southern Taiwan, 2012 to 2020: MLSA Phylogeny and Antimicrobial Susceptibility"

_antibiotics, 2022, doi:10.3390/antibiotics11101438_

Round 1

Reviewer 1 Report

Shu-Fang Kuo, Fang-Ju Chen, I-Chia Lan, Chun-Chih Chien and Chen-Hsiang Lee

Reviewer Comments:

The manuscript entitled “Epidemiology and Antimicrobial Susceptibility of Nocardia Species: A Large Tertiary Laboratory in Southern Taiwan, 2012 to 2020”, with Reference Submission “antibiotics-1952660” provides the identification and susceptibility analysis of a substantial population of 77 Nocardia strains submitted during 2012-2020 to a tertiary hospital in Taiwan, by MALDI-TOF, the Multilocus Sequence Analysis schema, 16S rRNA, gyrB, hsp65, secA and rpoB analysis, PFGE, and the broth microdilution method, respectively.

MAYOR COMMENTS

1.                   The title should be clarify

2.                   Regarding the shortcomings of the commercial MALDI-TOF MS database for the identification of Nocardia species, a species identification based on the complete sequence of the gold standard 16S rRNA is recommended to confirm species assigment.

3.                   Even though in the Introduction is indicated the employment of the MLSA and the sequencing of the gyrB, secA, hsp65 and/or rpoB for the identification of Nocardia species, the results are not showed in the text. In addition, the result and discussion of the 16S rRNA and MLSA schema are unclear and confusing. Moreover, Table 4 was mentioned in the text but it was not included.

4.                   A language assistant should be consulted for minor spell check

MINNOR COMMENTS

Please, ensure that the number of words in the text meet the criteria for the publication in this journal as an article.

Line 2: The title should be clarify

Line 20: Please, substitute “Sensititre RAPMYCO TREK” by “broth microdilution (BMD) method”, because it will be explained in the “Material and methods” section.

Line 23: Please, ensure that the “gyrB, secA, hsp65 and/or rpoB gene” analysis is included in the text, or remove this sentence.

Line 36: Please, consider the substitution of the reference 1 (Palomba, 2022), for an standard reference when you are explaining the transmission of nocardiosis, as Brown-Elliott et al., 2006, for example.

Line 53. The phylogenetic analysis of gyrB, secA, hsp65 and rpoB genes was not included in the text.

Line 58: Redundant data between text and table 1. Please, consider their removal in the text or the elimination of the Table 1.

Line 64. It is unclear if the 16S rRNA gene sequencing was applied only to these four strains or to the 77 strains. Please, consider the reformulation of the sentence “A total of 73 isolates were identified with MALDI-TOF, and the remaining four strains that could not be identified with MALDI-TOF were sequenced with 16S rRNA” by: “Seventy-seven Nocardia strains were identified, 73 by the MALDI-TOF MS platform with a score ≥ 2, and the remaining four strains that mismatched the level-species criteria for MALDI-TOF, were furthermore identified by the 16S rRNA analysis”. Please, clarify if the 16S rRNA sequencing was also applied to the remaining 73 strains previously identified by MALDI-TOF.

Line 67: Please, change “ateroides” by “asteroides” and “concave” by “concava”.

Line 71: This paragraph sounds redundant. Please consider substitute “Compared with the drug pattern types described by McTaggart [9], we found that both N. farcinica and N. cyriacigeorgica in our study were IPM-resistant and N. cyriacigeorgica was resistant, unlike a previous study showing that N. farcinica and N. cyriacigeorgica were susceptible to IPM and N. cyriacigeorgica was susceptible to FEP” by “In contrast to the drug pattern types described previously by McTaggart et al. [9], we found that both N. farcinica and N. cyriacigeorgica strains were IPM-resistant and N. cyriacigeorgica strains were as well FEP-resistant”.

Line 83: Please, check the verb.

Line 85: Figure 2 also showed that the prevalence in autumn was higher. Please, include it in the text.

Line 88: The results should be better included in the text, not in the footnote.

Line 93: The results should be better included in the text.

Line 102: Please, include “The susceptibility breakpoints for tigecycline and cefoxitin are not stablished”

Line 104 (Table 3): Please, ensure that this table meet the criteria for the publication in this journal, even considering the landscape format.

The percentage of 0% or 100% when it is only one strains (N. niigatensis, N. otitidiscaviarum, N. crassostreae, N. cerradoensis and N. amikacinitolerans) should be included as well.

Please, substitute “N. ateroides” by “N. asteorides”.

Line 108: Table 4 was not included in the text.

Line 117: The results should be included in the text.

Line 119: Please substitute “16. S rRNA” by “16S rRNA”

Line 121: This section is unclear. Does the “nucleic acid” refers to the MLSA concatenated sequence? Is the Figure 4 based on 16S rRNA or on the MLSA schema? In addition, the phylogenetic tree represented in Figure 4 was constructed from a unique sequence of each species identified by MALDI-TOF. I strongly recommend the recalculation based on the 77 sequences of the studied strains to display de intraspecies clustering and the interspecies distance, both based on the 16S rRNA and MLSA concatenate sequence, and their corresponding analysis in the Discussion section.

Line 131: The results should be better included in the text.

Line 133: The result of the MLSA schema was not explained in the text

Line 136: Nowadays, the small number of MSs in the commercial Bruker database for unusual and recently described species renders the method unreliable for a correct identification in absence of an in-house database. In Conville et al, reference 8, it is explained: “Recently, MALDI-TOF MS has been shown to provide accurate identification of Nocardia species when an augmented Nocardia library is employed. However, while some species are easily identified (i.e., N. brasiliensis), for others, the identification has only been shown to extend to the complex level (N. abscessus complex, N. brevicatena-N. paucivorans complex, N. nova complex, and N. transvalensis complex) (12,–14). The identification of uncommon species remains a challenge. Variation in the spectral profiles exist for some Nocardia species (i.e., N. cyriacigeorgica), and it is unclear if these variations represent taxonomic heterogeneity“

However, this platform without supplemental spectra could be employed for the rapid identification of the most common Nocardia species.

Please, consider this idea and reformulate this sentence.

Line 137: Please, add a reference to this asseveration.

Line 144: Please, compare the distribution of Nocardia spp. with contemporary publications, as Wang et al. (2020; DOI 10.1128/spectrum.01560-21) or Wei et al. (2021; DOI: 10.1186/s12866-021-02412-x).

Line 165: Neither in reference 18 not in reference 20,  all the Nocardia strains were isolated in Madrid and Ontario, respectively. This is the city were the Microbiology National Centers are located, but the strains could be submitted from any hospital of the country. Please, remove the name of these cities.

Line 176: The probability of transmission among people of nocardiosis, and its incidence, are low, so I discourage the consideration of nocardiosis as a global problem, but a phase as “Nocardiosis occurs worldwide” can be added. It should be better highlight the escalating globally in the number of Nocardia infections, possibly in relation to the increased number of immunocompromised patients and improved laboratory techniques for nocardiosis detection, or the underestimation due to its uncertain clinical manifestations.

Line 179: The PFGE is not the only technique explored here for molecular typing, could you please compare these results in N. cyriacigeorgica with those obtained by MLST? This recommendation is based in previous studies (Schlaberg et al.), when the heterogeneity of both the hsp65 and 16S rRNA genes of isolates belonging to the species N. cyriacigeorgica were described.

Line 184: Even though Nocardia genus is considered an environmental pathogen, the comparison of the prevalence of Nocardia species per month should be better calculated among clinical samples, avoiding the soil samples.

Line 195: Please, include the number of studied strains

Line 200: In order to facilitate the understanding of the text, please, rename and reorder the Material and Methods sections as in the Results section, being then the point 4.2 renamed as “Identification of Nocardia species”, including MALDI-TOF MS procedure, 16S rRNA and MLSA sequencing, and the identification based on the susceptibility patterns stablished by Wallace et al.

Line 201: Please, detail the protein extraction method employed for the MALTI-TOF MS identification (maybe the boiling method), adding the corresponding reference.

Line 207: Please, change “16s” by “16S”

Line 202: Could you please clarify if the rpoB gene was sequenced?

Line 211: Discrepancies between short and full 16S gene sequences analysis have been described, and the discrimination grows when the length increases (“All Nocardia species include a variable region near the 5′ terminus of the gene, which allows for the differentiation of the majority of the currently recognized Nocardia species with a partial (500 bp) 16S rRNA gene sequence. Extended sequencing of the entire gene may be required for the recognition of a new species or for the discrimination of closely related species that have identical or nearly identical, sequences within the first 500 bp of the gene” (Conville et al., 2017)

It is recommended the employment of the full-length 16S rRNA sequence for species assignment.

Moreover, the genes sequences studied (16S rRNA, gyrB, secA, hsp65 and/or rpoB) should be deposited in the GenBank databases and their correspondig accession numbers added to the text.

Line 217: The corresponding reference should be McTaggart et al. (10).

Line 218: Please, add the corresponding reference for the NJ method.

Line 226: Could you please explain the election of the nine N. cyriacigeorgica strains for the PFGE analysis? Due to the low number of strains, it would be interesting the study in the 25 N. cyriacigeorgica strains identified by MALDI—TOF MS.

Line 232: Could you please include the platform employed for the analysis and graphics of the PFGE?

Line 224: This could be relocated in the “Identification of Nocardia species” section

Line 249: Please, rephrase this sentence to settle the significance.

Supplementary File: Please add the references and review the PCR cycling conditions

Author Response

MAYOR COMMENTS

 The title should be clarify

Response: Thank you for your suggestion. We clarify the title as “Epidemiology of Nocardia Species at a Tertiary Hospital in Southern Taiwan, 2012 to 2020: MLSA Phylogeny and Antimicrobial Susceptibility”.

  1. Regarding the shortcomings of the commercial MALDI-TOF MS database for the identification of Nocardiaspecies, a species identification based on the complete sequence of the gold standard 16S rRNA is recommended to confirm species assigment.

Response: Thank you for your comment. 16S rRNA sequencing is the gold standard method for species identification of Nocardia species. Total 77 Nocardia isolates in this study were collected and identified to species level using multi-locus sequence analysis (MLSA). Among these isolates, 12 type strains were identified under a phylogenic tree contracted from the concatenated gyrB-16S rRNA-secA1-hsp65 sequences. The gene sequences were deposited in the GenBank database and their corresponding accession numbers (Table S1). We also add the limitation of MALDI-TOF MS for identification of Nocardia species in the discussion. (Line 145)

  1. Even though in the Introduction is indicated the employment of the MLSA and the sequencing of the gyrBsecAhsp65and/or rpoB for the identification of Nocardia species, the results are not showed in the text. In addition, the result and discussion of the 16S rRNA and MLSA schema are unclear and confusing. Moreover, Table 4 was mentioned in the text but it was not included.

Response: Thank you for your reminder. It should be Table 3, not Table 4. We correct this error. We also rewrite the result of Phylogenetic tree by MLSA scheme. (Line 127) The limitation of MALDI-TOF MS for identification of Nocardia species has been put in the discussion. (Line 145)

  1. A language assistant should be consulted for minor spell check

Response: Thank you for the comment. This article has been revised by English writers. The certification is attached below.

MINNOR COMMENTS

Please, ensure that the number of words in the text meet the criteria for the publication in this journal as an article.

Line 2: The title should be clarify

Response: Thank you for your suggestion. We clarify the title as “Epidemiology of Nocardia Species at a Tertiary Hospital in Southern Taiwan, 2012 to 2020: MLSA Phylogeny and Antimicrobial Susceptibility”.

Line 20: Please, substitute “Sensititre RAPMYCO TREK” by “broth microdilution (BMD) method”, because it will be explained in the “Material and methods” section.

Response: Thank you for your suggestion. We replace the “Sensititre RAPMYCO TREK” with “broth microdilution method”. (Line 22)

Line 23: Please, ensure that the “gyrBsecAhsp65 and/or rpoB gene” analysis is included in the text, or remove this sentence.

Response: Thank you for your suggestion. We remove this sentence.

Line 36: Please, consider the substitution of the reference 1 (Palomba, 2022), for an standard reference when you are explaining the transmission of nocardiosis, as Brown-Elliott et al., 2006, for example.

Response: Thank you for your suggestion. We rewrite the reference 1 as Brown-Elliott et al., 2006.

Line 53. The phylogenetic analysis of gyrBsecAhsp65 and rpoB genes was not included in the text.

Response: Thank you for your reminder. We remove this sentence.

Line 58: Redundant data between text and table 1. Please, consider their removal in the text or the elimination of the Table 1.

Response: Thank you for your suggestion. We rewrite the sentence as “The clinical characteristics of the 77 Nocardia isolates were shown in Table 1.” (Line 59)

Line 64. It is unclear if the 16S rRNA gene sequencing was applied only to these four strains or to the 77 strains. Please, consider the reformulation of the sentence “A total of 73 isolates were identified with MALDI-TOF, and the remaining four strains that could not be identified with MALDI-TOF were sequenced with 16S rRNA” by: “Seventy-seven Nocardia strains were identified, 73 by the MALDI-TOF MS platform with a score ≥ 2, and the remaining four strains that mismatched the level-species criteria for MALDI-TOF, were furthermore identified by the 16S rRNA analysis”. Please, clarify if the 16S rRNA sequencing was also applied to the remaining 73 strains previously identified by MALDI-TOF.

Response: Thank you for your suggestion. We rewrite the material and method for species level identification using multi-locus sequence analysis (MLSA). As a result, we rewrite the phrase as “Of the 77 Nocardia isolates, 12 type strains were identified under a phylogenetic tree constructed from the concatenated gyrB-16S rRNA-secA1-hsp65 sequences.” (Line 67)

Line 67: Please, change “ateroides” by “asteroides” and “concave” by “concava”.

Response: Thank you for your reminder. We change “ateroides” by “asteroides” and “concave” by “concava” in the revised edition.

Line 71: This paragraph sounds redundant. Please consider substitute “Compared with the drug pattern types described by McTaggart [9], we found that both N. farcinica and N. cyriacigeorgica in our study were IPM-resistant and N. cyriacigeorgica was resistant, unlike a previous study showing that N. farcinica and N. cyriacigeorgica were susceptible to IPM and N. cyriacigeorgica was susceptible to FEP” by “In contrast to the drug pattern types described previously by McTaggart et al. [9], we found that both N. farcinica and N. cyriacigeorgica strains were IPM-resistant and N. cyriacigeorgica strains were as well FEP-resistant”.

Response: Thank you for your suggestion. We rewrite the sentence as your suggestion. (Line 75)

Line 83: Please, check the verb.

Response: Thank you for your reminder. The verb has been checked. (Line 89)

Line 85: Figure 2 also showed that the prevalence in autumn was higher. Please, include it in the text.

Response: Thank you for your suggestion. We rewrite the sentence as “the prevalence of Nocardia infections was lower in summer and higher in autumn.” (Line 92)

Line 88: The results should be better included in the text, not in the footnote.

Response: Thank you for your reminder. We delete the footnote.

Line 93: The results should be better included in the text.

Response: Thank you for your reminder. We delete the footnote.

Line 102: Please, include “The susceptibility breakpoints for tigecycline and cefoxitin are not stablished”

Response: Thank you for your suggestion. We add this sentence in the revised edition. (Line 108)

Line 104 (Table 3): Please, ensure that this table meet the criteria for the publication in this journal, even considering the landscape format.

The percentage of 0% or 100% when it is only one strains (N. niigatensis, N. otitidiscaviarum, N. crassostreae, N. cerradoensis and N. amikacinitolerans) should be included as well.

Please, substitute “N. ateroides” by “N. asteorides”.

Response: Thank you for your suggestion. We rewrite the Table 3 as your suggestion and change “ateroides” by “asteroides”.

Line 108: Table 4 was not included in the text.

Response: Thank you for your reminder. It should be Table 3, not Table 4. We correct this error.

Line 117: The results should be included in the text.

Response: Thank you for your reminder. We delete the footnote.

Line 119: Please substitute “16. S rRNA” by “16S rRNA”

Response: Thank you for your reminder. The 2.6 subtitle has been changed as “Phylogenetic tree by MLSA scheme”. (Line 127)

Line 121: This section is unclear. Does the “nucleic acid” refers to the MLSA concatenated sequence? Is the Figure 4 based on 16S rRNA or on the MLSA schema? In addition, the phylogenetic tree represented in Figure 4 was constructed from a unique sequence of each species identified by MALDI-TOF. I strongly recommend the recalculation based on the 77 sequences of the studied strains to display de intraspecies clustering and the interspecies distance, both based on the 16S rRNA and MLSA concatenate sequence, and their corresponding analysis in the Discussion section.

Response: Thank you for your comment. We rewrite the sentence as “In our study, there were 12 Nocardia type strains. The differences in four-locus (gyrB-16S rRNA-secA1-hsp65) MLSA concatenated sequences among these 12 species and the evolutionary phylogenetic trees are shown in Figure 4.” (Line 128)

Line 131: The results should be better included in the text.

Response: Thank you for your reminder. We delete the footnote.

Line 133: The result of the MLSA schema was not explained in the text

Response: Thank you for your suggestion. The result of phylogenic tree by MSA scheme was put in Line 127-136.

Line 136: Nowadays, the small number of MSs in the commercial Bruker database for unusual and recently described species renders the method unreliable for a correct identification in absence of an in-house database. In Conville et al, reference 8, it is explained: “Recently, MALDI-TOF MS has been shown to provide accurate identification of Nocardia species when an augmented Nocardia library is employed. However, while some species are easily identified (i.e., N. brasiliensis), for others, the identification has only been shown to extend to the complex level (N. abscessus complex, N. brevicatena-N. paucivorans complex, N. nova complex, and N. transvalensis complex) (12,–14). The identification of uncommon species remains a challenge. Variation in the spectral profiles exist for some Nocardia species (i.e., N. cyriacigeorgica), and it is unclear if these variations represent taxonomic heterogeneity“

However, this platform without supplemental spectra could be employed for the rapid identification of the most common Nocardia species.

Please, consider this idea and reformulate this sentence.

Response: Thank you for your suggestion. We rewrite the sentence as “16S rRNA sequencing is the gold standard method for species identification. Recently, MALDI-TOF MS has been shown to provide accurate identification of Nocardia species when an augmented Nocardia library is employed. However, while some species are easily identified (i.e., N. brasiliensis), for others, the identification has only been shown to extend to the complex level (N. abscessus complex, N. brevicatena-N. paucivorans complex, N. nova complex, and N. transvalensis complex). The identification of uncommon species remains a challenge. [7, 9].” (Line 145)

Line 137: Please, add a reference to this asseveration.

Response: Thank you for your comment. We add “Wei, M.; Xu, X.; Yang, J.; Wang, P.; Liu Y, Wang, S.; Yang, C.; Gu, L. MLSA phylogeny and antimicrobial susceptibility of clinical Nocardia isolates: a multicenter retrospective study in China. BMC Microbiol. 2021, 21, 342.” as the reference. (Line 153)

Line 144: Please, compare the distribution of Nocardia spp. with contemporary publications, as Wang et al. (2020; DOI 10.1128/spectrum.01560-21) or Wei et al. (2021; DOI: 10.1186/s12866-021-02412-x).

Response: Thank you for your suggestion. We rewrite the sentence as “Previous studies conducted before 2010 indicated that the most common Nocardia spp. in Taiwan was N. brasiliensis [11, 12]. In contrast, N. farcinica was the most common isolated species in China, 2009 to 2021 [13]. N. nova complex organisms were the most common isolates in the United States and Canada before 2008 [14, 15], and N. cyriacigeorgica was the most common pathogen in Spain before 2008 [16]. N. cyriacigeorgica as the most common causative agent of pulmonary nocardiosis in southern Taiwan, 2004 to 2010 [17] and China, 2010 to 2020 which pulmonary nocardiosis (90.2%) was the most common clinical presentation of infection [10].” (Line 159) The references “Wang et al. (2020; DOI 10.1128/spectrum.01560-21) and Wei et al. (2021; DOI: 10.1186/s12866-021-02412-x) “ are added.

Line 165: Neither in reference 18 not in reference 20,  all the Nocardia strains were isolated in Madrid and Ontario, respectively. This is the city were the Microbiology National Centers are located, but the strains could be submitted from any hospital of the country. Please, remove the name of these cities.

Response: Thank you for your comment. We remove the name of these cities.

Line 176: The probability of transmission among people of nocardiosis, and its incidence, are low, so I discourage the consideration of nocardiosis as a global problem, but a phase as “Nocardiosis occurs worldwide” can be added. It should be better highlight the escalating globally in the number of Nocardia infections, possibly in relation to the increased number of immunocompromised patients and improved laboratory techniques for nocardiosis detection, or the underestimation due to its uncertain clinical manifestations.

Response: Thank you for your comment. We rewrite the sentence as “Nocardiosis occurs worldwide. Nocardia infections have increased in the last decades, likely due to improved detection and identification methods and the expanding immunocompromised population [3].” (Line 194)

Line 179: The PFGE is not the only technique explored here for molecular typing, could you please compare these results in N. cyriacigeorgica with those obtained by MLST? This recommendation is based in previous studies (Schlaberg et al.), when the heterogeneity of both the hsp65 and 16S rRNA genes of isolates belonging to the species N. cyriacigeorgica were described.

Response: We agree your point that PFGE is not the only technique explored for molecular typing. N. cyriacigeorgica has been reported to caused outbreak. We performed a PFGE analysis to exclude suspected cluster of N. cyriacigeorgica infection. In an addition, the MLSA pattern is similar among these N. cyriacigeorgica.

Line 184: Even though Nocardia genus is considered an environmental pathogen, the comparison of the prevalence of Nocardia species per month should be better calculated among clinical samples, avoiding the soil samples.

Response: Thank you for your suggestion. We rewrite the sentence as “However, more studies of prevalence of Nocardia species among clinical samples per month are needed to gain insight into the correlation of climate change and distribution of Nocardia spp..” (Line 205)

Line 195: Please, include the number of studied strains

Response: Thank you for your suggestion. We add the number of studied strain in the revised edition. (Line 216)

Line 200: In order to facilitate the understanding of the text, please, rename and reorder the Material and Methods sections as in the Results section, being then the point 4.2 renamed as “Identification of Nocardia species”, including MALDI-TOF MS procedure, 16S rRNA and MLSA sequencing, and the identification based on the susceptibility patterns stablished by Wallace et al.

Response: Thank you for your comment. The 4.2 MALDI-TOPF MS identification of Nocardia spp. has been deleted.

Line 201: Please, detail the protein extraction method employed for the MALTI-TOF MS identification (maybe the boiling method), adding the corresponding reference.

Response: Thank you for your comment. The 4.2 MALDI-TOPF MS identification of Nocardia spp. has been deleted.

Line 207: Please, change “16s” by “16S”

Response: Thank you for your reminder. We change “16s” by “16S”. (Line 228)

Line 202: Could you please clarify if the rpoB gene was sequenced?

Response: Thank you for your reminder. The rpoB gene was not sequenced in this study.

Line 211: Discrepancies between short and full 16S gene sequences analysis have been described, and the discrimination grows when the length increases (“All Nocardia species include a variable region near the 5′ terminus of the gene, which allows for the differentiation of the majority of the currently recognized Nocardia species with a partial (500 bp) 16S rRNA gene sequence. Extended sequencing of the entire gene may be required for the recognition of a new species or for the discrimination of closely related species that have identical or nearly identical, sequences within the first 500 bp of the gene” (Conville et al., 2017)

It is recommended the employment of the full-length 16S rRNA sequence for species assignment.

Moreover, the genes sequences studied (16S rRNA, gyrB, secA, hsp65 and/or rpoB) should be deposited in the GenBank databases and their corresponding accession numbers added to the text.

Response: Thank you for your comment. The employment of the full-length 16S rRNA sequence for species assignment that the discrimination grows when the length increases. We performed 16S rRNA sequence according to the references of McTaggart et al., 2010 and Gnanam et al., 2020. It is considered that short 16S gene sequence analysis is sufficient. The gene sequences were submitted to the National Center for Biotechnology Information (NCBI) database and accession numbers were obtained (Table S1).

Line 217: The corresponding reference should be McTaggart et al. (10).

Response: Thank you for your suggestion. We rewrite the sentence as “Primer sequences published by McTaggart et al. [31] are presented in Table S2.” (Line 239)

Line 218: Please, add the corresponding reference for the NJ method.

Response: Thank you for your comment. We add “Gascuel, O. BIONJ: an improved version of the NJ algorithm based on a simple model of sequence data. Mol. Biol. Evol. 1997, 14, 685–695.” as a reference for the NJ method. (Line 244)

Line 226: Could you please explain the election of the nine N. cyriacigeorgica strains for the PFGE analysis? Due to the low number of strains, it would be interesting the study in the 25 N. cyriacigeorgica strains identified by MALDI—TOF MS.

Response: Thank you for your comment. We randomly selected nine N. cyriacigeorgica strains which was isolated at 2019 and 2020 for PFGE analysis to determine the genetic relatedness. (Line 118)

Line 232: Could you please include the platform employed for the analysis and graphics of the PFGE?

Response: Thank you for your reminder. We add the sentence as “The Dice coefficients of the PFGE profiles were analyzed with an UPGMA dendrogram using GelCompar II version 6.6.11 (Applied Maths BVBA, Kortrijk, Belgium).” to explain the platform employed for the analysis and graphics of the PFGE. (Line 255)

Line 224: This could be relocated in the “Identification of Nocardia species” section

Response: Thank you for your comment. We add 4.6 Antimicrobial susceptibility patterns as a new sub-title. (Line271)

Line 249: Please, rephrase this sentence to settle the significance.

Response: Thank you for your comment. We rewrite the sentence as “McTaggart et al. [8] suggested numerous rarely-occurring species using broth microdilution and divided them into four other drug patterns.” (Line 277)

Supplementary File: Please add the references and review the PCR cycling conditions

Response: Thank you for your suggestion. The Table S2 show the housekeeping genes primer and PCR cycling condition. We add “McTaggart, L.R.; Richardson, S.E.; Witkowska, M.; Zhang, S.X. Phylogeny and identification of Nocardia species on the basis of multilocus sequence analysis. J. Clin. Microbiol. 2010, 48, 4525–4533.” as the reference in the footnote.

Reviewer 2 Report

It would be interesting to know what type of pathology these species produces, since it is different to treat a mycetoma to a pulmonary nocardiosis or brain abscesses

ask the authors if they can add a table with the type of disease that causes Nocardia. Mycetoma, Pulmonary nocardiosis, brain abscesses, cutaneous nocardiosis. These data are part of the epidemiology

They only mention the type of sample. 

Author Response

It would be interesting to know what type of pathology these species produces, since it is different to treat a mycetoma to a pulmonary nocardiosis or brain abscesses

ask the authors if they can add a table with the type of disease that causes Nocardia. Mycetoma, Pulmonary nocardiosis, brain abscesses, cutaneous nocardiosis. These data are part of the epidemiology

They only mention the type of sample. 

Response: Thank you for your suggestion. We add the “site of involvement” in the Table 1 for the type of disease that causes nocardiosis.

Round 2

Reviewer 1 Report

The aim of the work was to evaluate the diversity of Nocardia species with different molecular analyses and the susceptibilities to different types of antimicrobials. An extensive work was carried out regarding the susceptibility test part, which represent the strength of this work. While, the weakness of this work is represented by the molecular analysis. The quality of the figures and tables is satisfactory and the statistical methods are valid and correctly applied. The manuscript has become clearer. It is important to recognize that the authors have made a great effort to improve the manuscript during the review process, kindly applying the most of the comments.

As the authors note in their new revision in line 146, 16S rRNA is the gold standard for the identification of Nocardia species. However, when the identification to species level is based on the partial 5ʹ 16S rDNA sequencing, as is this case, a second genetic loci, as the secA1 gene for isolate identification is recommended because 16S rDNA sequence analysis alone provides insufficient species-level resolution for many Nocardia spp., whereas secA1 gene sequence analysis is more discriminatory and gives better resolution to species level (Tan, 2020:DOI 10.1016/j.jgar.2019.06.018). Nevertheless, both genes where included in the MLSA schema employed for the species assignation.

My reservation arises when the Nocardia type strains sequences are selected to be representative of each species identified by MLSA (Figure 4). In line 68 “Of the 77 Nocardia isolates, 12 species were identified” was previously correct, because your analysis is not based on type strains from the databases, is based on your studied strains. In a phylogenetic tree constructed from the approximately 2000-bp concatenated gyrB-16S rRNA-secA1-hsp65 sequences of 77 clinical Nocardia isolates is very improbable the presence of only 12 haplotypes, even more, corresponding to those of the type strains present in the GenBank database. The figure 4 should be substituted by a phylogenetic neighbor-joining tree including these 12 type strains as an indicator and the 77 clinical strains studied in this nine years, based on the MLSA concatenated sequence. A similar tree to the Figure 1 in “MLSA phylogeny and antimicrobial susceptibility of clinical Nocardia isolates: a multicenter retrospective study in China; Ming Wei, 2021” or Figure 3b in “Molecular identification of Nocardia species causing endophthalmitis using multilocus sequence analysis (MLSA): a 10-year perspective; Hariharan Gnanam, 2021. The comparison of the MLSA schema of 12 type strains is not original.

Moreover are missing the GenBank Accession numbers of the 16S rRNA, gyrB, secA1 and hsp65 sequences of the 77 strains explored in this study, because table S1 only include those from the type strains. Please, submit the four genes sequence of the 77 strains to the NCBI, adding their corresponding accession number to the Table S1. Please, comment these results in the discussion section.

Minnor comments:

Line 83: The Table 2 employed bold type for “N. farcinica” and “18”. Please, employ this type in the header only, applying it to “Drug patterns types” and “Antimicrobial susceptibility pattern”.

Line 90: Please, change “are” by “was”, it is singular.

Line 91: Please, do not change “was” by “is”

Line 129: Please, do not change “spp.” by “type strains”. Could you please suggest a breakpoint in the MLSA schema similarity among the studied strains and the type strains for the species assignation?

Line 143: Figure 4 is duplicated

Line 165: Please, change “2008” by “2014”, is the period studied in this reference (from 2005 to 2014)

Line 252: The sequence of the 16S rRNA Nocardia –E8F primer do not correspond to this described in the table S1. Please, confirm the sequence of nucleotides and remove it from the text, it is repeated in table S1.

Please, also include the sequencing of the MLSA schema in the identification method (line 240-242), because in the reviewed version the strains were identified by MLSA, not 16S rRNA target.

Line 275: Please, remove “This may be useful for identifying species without MALDI-TOF or 16S rRNA gene sequencing” from the Material and Methods section

Author Response

Manuscript ID: antibiotics-1952660
Type of manuscript: Article
Title: Epidemiology and antimicrobial susceptibility of Nocardia species: a large tertiary laboratory in southern Taiwan, 2012 to 2020
Authors: Shu-Fang Kuo, Fang-Ju Chen, I-Chia Lan, Chun-Chih Chien, Chen-Hsiang Lee*

Response to Reviewer 1 Comments

Dear reviewer,

We are thankful for your thoughtful insights and suggestions. The manuscript has been benefited from these perceptive suggestions.

We have revised the manuscript and made all the necessary changes in accordance with reviewers’ suggestions. The changes are revised as “Track Changes”. We have also prepared a point-by-point response to the reviewers’ comments.

Thank you for your consideration. I am looking forward to hearing from you soon.

Sincerely,

Dr. Chen-Hsiang Lee

Division of Infection Diseases, Department of Internal Medicine, Kaohsiung Chang Gung Memorial Hospital, 123 Ta-Pei Road, Niao-Sung District, Kaohsiung 833, Taiwan.

TEL: +886-7-7317123 ext. 8304

FAX: +886-7-7-7322402

E-mail: lee900@cgmh.org.tw

The aim of the work was to evaluate the diversity of Nocardia species with different molecular analyses and the susceptibilities to different types of antimicrobials. An extensive work was carried out regarding the susceptibility test part, which represent the strength of this work. While, the weakness of this work is represented by the molecular analysis. The quality of the figures and tables is satisfactory and the statistical methods are valid and correctly applied. The manuscript has become clearer. It is important to recognize that the authors have made a great effort to improve the manuscript during the review process, kindly applying the most of the comments.

 Response: Thank you for your comment.

As the authors note in their new revision in line 146, 16S rRNA is the gold standard for the identification of Nocardia species. However, when the identification to species level is based on the partial 5ʹ 16S rDNA sequencing, as is this case, a second genetic loci, as the secA1 gene for isolate identification is recommended because 16S rDNA sequence analysis alone provides insufficient species-level resolution for many Nocardia spp., whereas secA1 gene sequence analysis is more discriminatory and gives better resolution to species level (Tan, 2020:DOI 10.1016/j.jgar.2019.06.018). Nevertheless, both genes where included in the MLSA schema employed for the species assignation.

 Response: Thank you for your comment. We agree with your point. We rewrite the phrase as “Sequence analysis of the 16S rRNA gene is suggested as the “gold standard” for the identification of Nocardia isolates to species level. However, when the identification to species level is based on the partial 5ʹ 16S rRNA sequencing, as is this case, a second genetic locus, as the secA1 gene for isolate identification is recommended because 16S rRNA sequence analysis alone provides insufficient species-level resolution for many Nocardia spp., whereas secA1 gene sequence analysis is more discriminatory and gives better resolution to species level [10]. Both genes where included in the MLSA schema employed in this study for the species assignation to achieve higher accuracy and differentiation.” (Line 133-141)

My reservation arises when the Nocardia type strains sequences are selected to be representative of each species identified by MLSA (Figure 4). In line 68 “Of the 77 Nocardia isolates, 12 species were identified” was previously correct, because your analysis is not based on type strains from the databases, is based on your studied strains. In a phylogenetic tree constructed from the approximately 2000-bp concatenated gyrB-16S rRNA-secA1-hsp65 sequences of 77 clinical Nocardia isolates is very improbable the presence of only 12 haplotypes, even more, corresponding to those of the type strains present in the GenBank database. The figure 4 should be substituted by a phylogenetic neighbor-joining tree including these 12 type strains as an indicator and the 77 clinical strains studied in this nine years, based on the MLSA concatenated sequence. A similar tree to the Figure 1 in “MLSA phylogeny and antimicrobial susceptibility of clinical Nocardia isolates: a multicenter retrospective study in China; Ming Wei, 2021” or Figure 3b in “Molecular identification of Nocardia species causing endophthalmitis using multilocus sequence analysis (MLSA): a 10-year perspective; Hariharan Gnanam, 2021. The comparison of the MLSA schema of 12 type strains is not original.

Response: Thank you for your comment. The Nocardia type strains sequences were selected to be representative of each species identified by MLSA (Figure 4). The Figure 4 has been substituted as “A phylogenetic neighbor-joining tree including these 12 type strains as an indicator and the 77 clinical strains studied in this nine years, based on the MLSA concatenated sequence.”

Moreover, are missing the GenBank Accession numbers of the 16S rRNA, gyrBsecA1 and hsp65 sequences of the 77 strains explored in this study, because table S1 only include those from the type strains. Please, submit the four genes sequence of the 77 strains to the NCBI, adding their corresponding accession number to the Table S1. Please, comment these results in the discussion section.

Response: Thank you for your comment. The Table S1 has been substituted as “Accession number for four genes of these 12 type strains of Nocardia species as an indicator from GenBank.”

Minor comments:

Line 83: The Table 2 employed bold type for “N. farcinica” and “18”. Please, employ this type in the header only, applying it to “Drug patterns types” and “Antimicrobial susceptibility pattern”.

Response: The change has been made. (Line 71)

Line 90: Please, change “are” by “was”, it is singular.

Response: The change has been made. (Line 77)

Line 91: Please, do not change “was” by “is”

Response: The change has been made. (Line 78)

Line 129: Please, do not change “spp.” by “type strains”. Could you please suggest a breakpoint in the MLSA schema similarity among the studied strains and the type strains for the species assignation?

Response: Thank you for your comment. We keep “spp.” in the revised edition. (Line 112)

Line 143: Figure 4 is duplicated

Response: We delete the duplicated Figure 4.

Line 165: Please, change “2008” by “2014”, is the period studied in this reference (from 2005 to 2014)

Response: To be clarified, we rewrite the sentence as “N. nova complex organisms were the most common isolates in the United States before 2004 and Canada before 2008”. (Line 150-151)

Line 252: The sequence of the 16S rRNA Nocardia –E8F primer do not correspond to this described in the table S1. Please, confirm the sequence of nucleotides and remove it from the text, it is repeated in table S1.

Please, also include the sequencing of the MLSA schema in the identification method (line 240-242), because in the reviewed version the strains were identified by MLSA, not 16S rRNA target.

Response: Thank you for your comment. The sentence of the sequence of the 16S rRNA Nocardia has been deleted. (Line 209-211)

Line 275: Please, remove “This may be useful for identifying species without MALDI-TOF or 16S rRNA gene sequencing” from the Material and Methods section

Response: The sentence “This may be useful for identifying species without MALDI-TOF or 16S rRNA gene sequencing” has been deleted. (Line 253-254)
